# Diversity and evolution of a phase-variable multi-locus antigen in *Neisseria gonorrhoeae*

QinQin Yu[1,2], Tatum D. Mortimer[3], Sofia O. P. Blomqvist[1,2], Bailey Bowcutt[1,2], David Helekal[1,2], Samantha G. Palace[1,2], Yonatan H. Grad[1,2]*

1 Department of Immunology and Infectious Diseases, Harvard T. H. Chan School of Public Health, Boston, Massachusetts, United States of America, 2 Center for Communicable Disease Dynamics, Harvard T. H. Chan School of Public Health, Boston, Massachusetts, United States of America, 3 Department of Population Health, University of Georgia College of Veterinary Medicine, Athens, Georgia, United States of America

* ygrad@hpsh.harvard.edu

## Abstract

*Neisseria gonorrhoeae* is a sexually transmitted bacterial pathogen that deploys multiple mechanisms to evade the immune system, including rapid variation in surface antigens. One of the most abundant and diverse antigens is a member of the opacity-associated (Opa) family, which includes surface proteins that mediate gonococcal attachment to human receptors. Studies of Opa diversity and evolution have been limited by the inability of short-read sequencing to resolve the multiple copies of *opa* in each genome, preventing a comprehensive understanding of variation in gonococcal antigens for vaccine design and immunology studies. We assembled a dataset of 219 complete genomes from phylogenetically diverse clinical isolates using long-read sequencing and developed bioinformatics and phylogenetics tools to assess *opa* variation quantitatively. Each genome had on average 7 distinct *opa* alleles at 9–12 *opa* loci, and almost all isolates had at least one pair of identical or near-identical *opa* genes. Fewer *opa* genes were in frame (and thus inferred to be expressed) than expected by chance. While genomic distance between isolates correlated with overall *opa* allele sequence similarity, *opa* genes were on average 74 times more diverse than the rest of the genome. One *opa* locus evolved more rapidly than the other loci. There was little evidence that interspecies recombination contributed to *N. gonorrhoeae opa* diversity. Our findings reveal a continuously evolving *opa* repertoire that leads to diverse *opa* alleles even in closely related strains and indicate that there are likely unknown biological factors modulating *opa* reading frame.

## Author summary

The rising levels of antibiotic resistance in *Neisseria gonorrhoeae* make controlling the spread of this sexually transmitted pathogen a public health priority.

**Data availability statement:** The Nanopore sequencing reads and assembled complete genomes generated in this study are available at the European Nucleotide Archive under study accession PRJEB106623: http://www.ebi.ac.uk/ena/browser/view/PRJEB106623. Our code for performing the analyses is available in GitHub (https://github.com/qinqin-yu/gonococcus_opa_diversity).

**Funding:** This project received support from the American Sexually Transmitted Diseases Association (YHG and QY, https://www.astda.org/), National Institutes of Health R21 AI172369 (YHG, https://grants.nih.gov/), and National Institutes of Health T32 AI007535 (QY, https://grants.nih.gov/). The funders had no role in study design, data collection and analysis, decision to publish, or preparation of the manuscript.

**Competing interests:** The authors have declared that no competing interests exist.

*N. gonorrhoeae* rapidly varies surface proteins to evade recognition by the human adaptive immune system. Understanding how these proteins evolve may help us design better vaccines and control measures for curbing the spread of gonorrhea. One of the most abundant surface-exposed proteins is a member of the opacity-associated (Opa) family and help *N. gonorrhoeae* bind to host cells upon colonization. Research efforts to understand the evolution of Opa have been limited because it is encoded by multiple genes in the genome that are not resolved by short-read sequencing technologies. Here, we resolved the genes that encode Opa using a dataset of 132 publicly available complete genomes and 87 genomes that we completed using long-read sequencing of phylogenetically diverse clinical isolates. We found that Opa evolves rapidly to generate different versions of the protein in the same isolate, but very few of these protein versions are encoded by sequences that are in frame. We also found evidence that there may be other, uncharacterized mechanisms that control how these proteins evolve over longer timescales.

## Introduction

In 2020, the World Health Organization (WHO) estimated there were more than 82 million cases worldwide of gonorrhea, the sexually transmitted infection caused by the obligate human pathogen *Neisseria gonorrhoeae*. Gonorrhea is a WHO priority pathogen due to increasing levels of antibiotic resistance and no highly-effective vaccine [1]. Infection does not appear to confer protection, such that individuals can be reinfected [2], even by the same strain [3], reflecting *N. gonorrhoeae*'s escape from the host immune system [4,5].

One of the ways *N. gonorrhoeae* achieves this immune evasion is through rapidly varying its antigens [6,7]. A major family of surface-exposed proteins that undergo this variation are the opacity-associated (Opa) proteins [8–11]. Opa proteins mediate gonococcal attachment to human receptors during colonization [12–17] and may be involved in suppression of the host immune response, though the mechanisms are as yet unclear [15,18]. Opa proteins also mediate non-opsonic phagocytosis by human neutrophils [19–21]. These proteins are encoded by multiple highly diverse *opa* genes in separate loci in each *N. gonorrhoeae* genome [10,22]. A series of pentanucleotide repeats in the signal sequence-encoding region at the 5' end of the coding sequence determines whether each of these genes is in frame and fully translated, and changes in the repeat copy number, likely from slip-strand repair, result in phase variation [23].

Previous studies of *opa* variation and evolution have been limited to small numbers of isolates, often closely related [9,22,24]. These studies showed that *opa* genes have high sequence diversity within and between isolates [9], recombination within isolates plays a key role in generating new sequence diversity [11,24], and gene conversion events can lead to two identical or near-identical *opa* alleles [23–25]. Bilek et al. [24] observed that one *opa* gene varied more in closely related isolates than the

other *opa* loci. There is evidence of correlation between promoter strength and the rate of *opa* phase variation [26]. There is also evidence that *opa* loss occasionally occurs [24] and few *opa* are phase on in each isolate [23]. Phylogenetic analysis of *opa* from *N. gonorrhoeae, N. meningitidis,* and the commensals *N. flava* and *N. sicca* showed that *opa* clustered by species [9].

However, due to the small numbers of isolates in these studies, we lack generalizable quantitative patterns and thus have a limited understanding of *N. gonorrhoeae opa* sequence variation and evolution. Several obstacles have prevented studies of *opa* diversity in larger numbers of isolates. Polymerase chain reaction can amplify the *opa* regions [22,24], but this approach is challenging to scale up due to unknown primer coverage of the *opa* repertoire. *opa* sequences cannot be resolved using short-read whole-genome sequencing platforms, such as Illumina, as the multiple and highly similar *opa* loci are problematic for mapping and genome assembly. Furthermore, the high diversity of *opa* has made it challenging to analyze sequences using standard bioinformatics, phylogenetics, and phylodynamics approaches, in that methods are designed primarily for single copy genes.

Long-read sequencing enables us to overcome these challenges. We combined publicly available complete genomes with diverse genomes that we sequenced using the Oxford Nanopore long-read platform. We developed bioinformatics and phylogenetics approaches for defining the diversity and evolution of highly variable multi-locus antigens and applied them to the largest set of *opa* sequences to date to investigate *opa* variation. Finally, we developed a sequence-based clustering approach for *opa* alleles in the effort towards establishing a standardized nomenclature to facilitate comparison of *opa* alleles in future studies.

## Methods

### Dataset of high-quality complete genomes

**Publicly available complete genomes.** We searched for "*Neisseria gonorrhoeae*" in the National Center for Biotechnology Information (NCBI) Genome Datasets (https://www.ncbi.nlm.nih.gov/datasets/genome/) and downloaded all genomes with an assembly level of "chromosome" or "complete" on July 31, 2025. We determined the sequencing technology and assembly method from the associated publications (where available) and associated metadata available on NCBI. In the rare cases of discrepancies between the publication and NCBI, we used the metadata reported in the publication.

**Nanopore sequencing and genome assembly.** *N. gonorrhoeae* isolates were streaked from frozen stocks onto GCB agar (Difco) with Kellogg's supplement [27] and grown at 37° C with 5% $CO_2$ overnight. Genomic DNA was extracted using the Invitrogen PureLink Genomic DNA Mini Kit. We prepared the library using the Oxford Nanopore Native Barcoding Kit 24 V14 (SQK-NBD114.24) and performed Nanopore sequencing using a MinION Mk1B device. Basecalling was performed using Dorado v0.8.2 [28] superaccuracy basecalling. Reads were demultiplexed using Dorado v0.8.2 demux and filtered using Filtlong v0.2.1 (https://github.com/rrwick/Filtlong) to keep the 90% highest quality read bases and only reads longer than 1kbp. Genomes were assembled using Autocycler v0.2.1 [29] with 4 read subsets at 25x minimum depth and using the assemblers: Canu v2.2 [30], Flye v2.9.5 [31], miniasm v0.3 [32], NECAT v0.0.1 [33], NextDenovo v2.5.2 [34], and Raven v1.8.3 [35]. Genomes that were fully resolved (one complete genome plus optional plasmids) and had a length of at least 2 Mbp were kept, as we expect the length of the *N. gonorrhoeae* genome to be 2.2 Mbp.

### Identifying *opa* genes in complete genomes

Genomes were rotated using the rotate program v1.0 [36] to match the first 90 bases of *dnaA* from FA1090 (NC_002946.2), allowing for up to 5 mismatches. We wrote a custom script to identify *opa* genes that searched for the tandem pentanucleotide repeats (CTCTT) and a unique conserved sequence near the stop codon [22,24]. Details of the algorithm are presented in S1 File and S1 Fig. The N-terminus of the mature protein was found by searching for the

sequence GCAAGTGA [23], allowing at most 2 substitutions. Only sequences that have an identified start codon, stop codon, and N-terminus are determined to be intact *opa* genes. Intact *opa* genes were given a number starting from 1 in the order that they appeared in the rotated genome (i.e., FA1090 *opa1* is the first *opa* in FA1090 after *dnaA*). The *opa* sequence was determined to be in frame if the number of nucleotides between the start codon and the N-terminus was a multiple of 3.

**Comparison to other sequence search methods.** To confirm that we had identified all the *opa* genes in the genomes, we compared our method to three additional approaches:

1. We performed a search of the conserved region of the FA1090 *opa1* sequence between the semivariable and hypervariable 1 regions using BLAST v2.14.1. We set the maximum number of high-scoring segment pairs (local alignments) to keep as 15 for a single query-subject pair and kept all other parameters at the default values. We matched the hits from the BLAST results that were also found by our search algorithm by merging the nearest start positions of the hits from the two search algorithms if they were within 1200 nucleotides. BLAST hits that were not merged were identified as putative additional hits found by BLAST not found by our search algorithm and we manually inspected these sequences.

2. We annotated the genomes using Prokka v1.14.6 [37]. We set the genus to "Neisseria" and species to "gonorrhoeae" and kept all other parameters at their default values. We noticed that all the *opa* genes identified with our method had the gene name *piiC_\**, where * was a number. We then checked if there were any additional genes annotated as *piiC_\** that were not identified by our method.

3. We used Roary v3.13.0 [38] to define the pan-genome and cluster genes. We set the minimum percentage identity for the BLASTp program within Roary to be 90% and kept all other parameters at their default values. We looked at genes that clustered with the *opa* genes identified by our method to determine if there were any additional genes in these clusters not identified by our method.

**Identification of *opa* pseudogenes.** To identify *opa* pseudogenes that were missing the start and/or stop codon, we extracted the sequences, including upstream and downstream sequences, of the hits found by BLAST that were not found by our search algorithm and aligned the sequences with the FA1090 *opa* sequences using MAFFT v7.520 [39]. We visualized the alignments using JalView v2.11.5.1 [40] to identify if each hit was an *opa* pseudogene compared to the FA1090 *opa* sequences.

**Genomic rearrangements.** Genomic rearrangements were detected using progressiveMauve v2.4.0 [41] by comparing each complete genome (query) with the complete FA1090 genome (reference). We rearranged the *opa* position in the query genome relative to the reference genome by reordering and flipping regions of the query genome that fell into locally collinear blocks compared to the reference. Regions of the query genome that did not fall into any locally collinear block with the reference genome were not altered.

## Whole genome phylogenies

**Generating reference-mapped pseudogenomes.** To generate reference-mapped pseudogenomes, we used short reads generated by, or simulated to be generated by, Illumina sequencing. Sequencing reads from all publicly available *N. gonorrhoeae* datasets were downloaded from the European Nucleotide Archive (S3 Table and https://github.com/qinqin-yu/gonococcus_opa_diversity/blob/main/opa_diversity_snakemake/input_data/whole_genome_metadata/global_isolates_metadata_met_qc.csv). For the publicly available complete genomes, we used ART v2016.06.05 [42] to simulate paired-end short reads from the MiSeq v3 sequencing system with a read length of 250 bp, 80x coverage, mean DNA fragment size of 600 and standard deviation of 100. Reads were mapped to the NCCP11945 (NC_011035.1) reference genome using BWA-MEM v0.7.17 [43]. Duplicate reads were marked with Picard v3.0.0 (https://broadinstitute.github.io/picard/), and reads were

sorted with SAMtools v1.17 [44]. The quality of the mapped reads was assessed using Qualimap's bamqc v2.2.1 [45]. We used Pilon v1.24 [46] to call variants (minimum mapping quality 20 and minimum coverage of 10). To create pseudogenomes, we replaced the reference allele with high quality variant calls (at least 90% of reads supporting the allele). Positions called as deletions by Pilon were replaced with a gap character, and positions with low coverage or an indeterminate allele were replaced with an N.

**Selecting representative global genomes.**  To select representative global genomes from the publicly available *N. gonorrhoeae* isolates, we included isolates with genomes meeting the following quality control filters: at least 80% of reads mapped to the *N. gonorrhoeae* reference genome, the coverage of reads mapped to the *N. gonorrhoeae* reference genome was > 40X, fewer than 12% of the sites in the *N. gonorrhoeae* reference genome were unable to be confidently called using our assembly pipeline, and the *de novo* assembly length was 1.75 Mbp-2.5 Mbp. This yielded 21,653 genomes which were then clustered using PopPUNK v2.6.0 [47]. The isolate in each PopPUNK cluster with the fewest contigs in its *de novo* assembly was chosen as the representative isolate from that cluster, resulting in 737 representative genomes (S4 Table).

**Recombination-masked phylogenies.**  Recombination-masked phylogenetic trees were created using the reference-mapped pseudogenomes in Gubbins v3.3.4 [48] using the GTR substitution model, the RAxML Next Generation tree builder [49], a maximum of 20 iterations for the phylogeny of complete genomes, and a maximum of 5 iterations for the phylogeny of representative and complete genomes.

## Quantifying *opa* diversity

**Comparison of within- and between-isolate *opa* diversity.**  To compare within- and between-isolate *opa* diversity, we first translated the *opa* sequences downstream of the coding repeats. We excluded all sequences with premature stop codons due to frameshift mutations downstream of the coding repeats or nonsense mutations. We then aligned the amino acid sequences using MAFFT v7.520 [39] with the default parameters. We then calculated pairwise distances in the alignment for all pairs of *opa* sequences from different isolates (between-isolate comparison) and all pairs of *opa* sequences from the same isolate (within-isolate comparison).

**Similar *opa* within the same isolate.**  To identify clusters of similar *opa* sequences from the same strain, we used the NetworkX Python package (https://github.com/networkx/networkx) to create graphs where the nodes are the *opa* sequences and the edges connect *opa* sequences with less than 5% difference in amino acid sequence alignment. The clusters of similar *opa* sequences were identified as the connected components in the graph. The percentage of isolates with at least one pair of similar *opa* was determined by finding the isolates that contained at least one *opa* sequence with less than 5% difference in amino acid sequence alignment with another *opa* sequence in the same strain.

**Comparison of *opa* sequence distance and genomic sequence distance.**  We calculated the *opa* sequence distance and the genomic distance between pairs of genomes. To facilitate quantitative comparisons between pairs of genomes, only pairs of genomes with the same number of *opa* genes were kept. The *opa* sequence distance was calculated using the alignment generated above and pairing the *opa* sequences between the two genomes that had the fewest number of segregating sites using a greedy algorithm (i.e., find the two closest *opa* sequences, then find two next closest *opa* sequences, etc. If there are identical matches, then one is chosen at random to be matched.). The total *opa* sequence distance was calculated by summing the distances between each pair of *opa* sequences and dividing by the number of *opa* genes (where a distance of 0 indicates the *opa* repertoire in the two genomes are identical and a distance of 1 indicates that all sites in all *opa* genes in the two genomes are different).

The pairwise genomic sequence distance was calculated as the number of SNPs between pairs of pseudogenomes divided by the length of the pseudogenome using pairsnp v0.3.1 (https://github.com/gtonkinhill/pairsnp). Missing sites (indicated by N) were excluded in the calculation of the SNP distance. Sites that were missing in some isolates but present in the two isolates being compared were included.

## Other *Neisseria* species *opa*

**Identifying *opa* genes.** We downloaded complete genomes ("chromosome" or "complete" assembly levels) from NCBI RefSeq on February 13, 2025 using the search term "Neisseria" and filtered out all *N. gonorrhoeae* genomes (S5 Table). Genomes were rotated using the rotate program v1.0 [36] to match the first 90 bases of *dnaA* from FA1090 (NC_002946.2), allowing for up to 5 mismatches.

If this sequence was not found, then the unrotated genome was used for downstream analysis. This occurred in 60/198 (30%) genomes. We identified *opa* genes using the custom script that we used for *N. gonorrhoeae*, which looks for the coding repeats and the unique conserved sequence near the stop codon. To look for additional *opa* that were missed by our script, we BLASTed the conserved regions of *N. gonorrhoeae* FA1090 *opa1*.

**Assessing sequence relatedness.** We calculated the k-mer distances using MASH v2.3 [50] with a k-mer size of 9 and used the distances as input to create a neighbor joining tree using rapidNJ v2.3.2 [51].

## Clustering of variable region sequences

**Definition of semivariable, hypervariable 1, and hypervariable 2 region sequences.** We aligned the nucleotide sequences of all *opa* genes using MAFFT v7.520 [39] with a gap opening penalty of 4 and a gap extension penalty of 1. The nucleotide sequence of FA1090 *opa1* was compared to the sequences in Bhat et al. [22] to identify the semivariable, hypervariable 1, and hypervariable 2 region sequences. The variable regions in the other *opa* sequences were determined based on where they aligned to FA1090 *opa1*.

**Calculation of k-mer distance.** The k-mer distances between sequences in each of the semivariable, hypervariable 1, and hypervariable 2 regions were calculated using MASH v2.3 [50] with a k-mer size of 6 for the semivariable region and 7 for the hypervariable 1 and hypervariable 2 regions. These k-mer sizes were chosen so that the probability of observing a random sequence of length k in a random sequence the length of each region was 0.01. The p-value is the probability of observing a given k-mer distance (or less) under the null hypothesis that both sequences are random collections of k-mers.

**Clustering.** The variable region sequences were clustered using the Markov Cluster Algorithm (MCL) v14.137 [52]. We constructed a distance matrix using the -log10 transformed p-value from MASH and values larger than 200 were set to 200. We made these transformations following the approach in TRIBE-MCL, which clusters proteins using -log10 transformed e-values from BLAST [53]. The distance matrix was input into MCL, which then performs multiple rounds of expansion and contraction to amplify large values and reduce small values of the matrix. The number of rounds of expansion and contraction is set by the inflation parameter. We ran the clustering using a range of inflation parameters: 1.4, 2, 4, 6, 8, 10, 12, and 14. We chose the inflation parameter that gave a stable clustering, inspired by TRIBE-MCL. A stable clustering was defined as having a percentage difference in clustering with the next lowest inflation parameter as less than 1% for the semivariable and hypervariable 2 regions. For the hypervariable 1 region, we relaxed the definition for a stable clustering as having less than a 5% difference in clustering with the next lowest inflation parameter due to the large number of clusters. The percentage difference in clustering is calculated as the sum of the number of nodes (sequences) that must be exchanged to transform the two clusterings into each other (also called the split/join distance) divided by two times the number of sequences. The motivation for this approach is that TRIBE-MCL uses convergence of outputs to determine the inflation parameter.

The cluster sequence logos were visualized by aligning the nucleotide sequences in each cluster using MAFFT v7.520 [39] with the default parameters and plotting the sequence logos with Logomaker v0.8 [54].

## Ancestral state reconstruction

**Dated phylogeny.** Pseudogenomes were generated as described above but with the reference genome being one randomly selected genome from the subtree. Regions of the genome under recombination were detected using Gubbins

v3.3.4 [48] (GTR substitution model, RAxML Next Gen tree builder, 20 iterations) and masked. The recombination-masked alignment was used to build a dated phylogeny in BEAST2 v2.7.5 [55], for all isolates that had date of collection information. When the date was given as a year, we set the date to January 1 of that year. We used a GTR substitution model with gamma rate heterogeneity (4 categories), a strict clock model, a coalescent constant population, and 100,000,000 chain length. Chains were inspected to confirm good mixing (effective sample size greater than 200).

**Ancestral state reconstruction.** We performed ancestral state reconstruction for each of the loci and variable region cluster types separately on the dated phylogeny using PastML v1.9.49 [56] with the default parameters. Isolates that did not have an intact *opa* at a locus were not included for the inference at that locus. If there were no changes to the variable region cluster type at a locus then the average rate of cluster type changes was set to 0.

## Phylogenies

Mutation-scaled phylogenies were visualized in iTOL v6.9.1 [57] using midpoint rooting. The time-scaled phylogeny was visualized in FigTree v1.4.4 (http://tree.bio.ed.ac.uk/software/figtree/).

## Antibiotic resistance phenotypes and alleles

We reported minimum inhibitory concentrations when they were available for a complete genome from an associated publication. The antibiotic resistance alleles for all complete genomes were determined using the NG-STAR scheme [58] as implemented in pyngoST v1.1.2 [59].

## Statistical analyses and analysis pipeline

Statistical analyses and plotting were done in Python v3.13.1. Snakemake v8.25.5 [60] was used for creating a reproducible analysis pipeline.

## Results

### A phylogenetically diverse set of 219 complete genomes of *N. gonorrhoeae*

Because long-read only assemblies using older sequencing technologies have a higher error rate, we filtered the publicly available complete genome assemblies to include only those assembled using long-read and short-read data, which yielded 132 complete genomes (S1 Table). To complement the publicly available complete genomes of *N. gonorrhoeae*, we sequenced an additional 87 isolates from across the *N. gonorrhoeae* phylogeny using Oxford Nanopore long-read sequencing (Fig 1 and S2 Table). In total, this gave 219 complete genomes. The isolates spanned 5 continents and the years 1962–2024 (S2 Fig) and had a range of antibiotic resistance phenotypes and alleles (S6 Table).

Comparison of three long-read only assembly methods (see S1 File) on two isolates whose genomes differed by 6648 SNPs showed that Autocycler performed the best, with no changes to the *opa* sequences for read depths of 21-140x (S3 Fig and S1 File). Polishing the Autocycler assemblies with short-read data did not change the *opa* sequences at any read depth, showing that the long-read data was sufficient to reconstruct the *opa* sequences. Further testing on a larger set of 11 phylogenetically diverse isolates indicated that varying read depths from 50-125x resulted in differences in *opa* sequence for only 4 out of 119 *opa* sequences in long-read only assemblies with Autocycler (3 SNPs and one *opa* that was undetected or had large numbers of sequence differences; see S4 Fig and S1 File). The genomes that we sequenced had coverages between 44x and 305x with an average coverage of 141x, with 84/87 (96.6%) above the threshold of 50x that we tested and all genomes above the threshold of 21x (S5 Fig).

### Identification of *opa* genes

Across the 219 complete genomes, we identified 2,359 full length *opa* genes. To determine if we missed any *opa* sequences, we compared our approach to three standard strategies to identify homologous genes: BLAST, Prokka, and

PLOS Pathogens

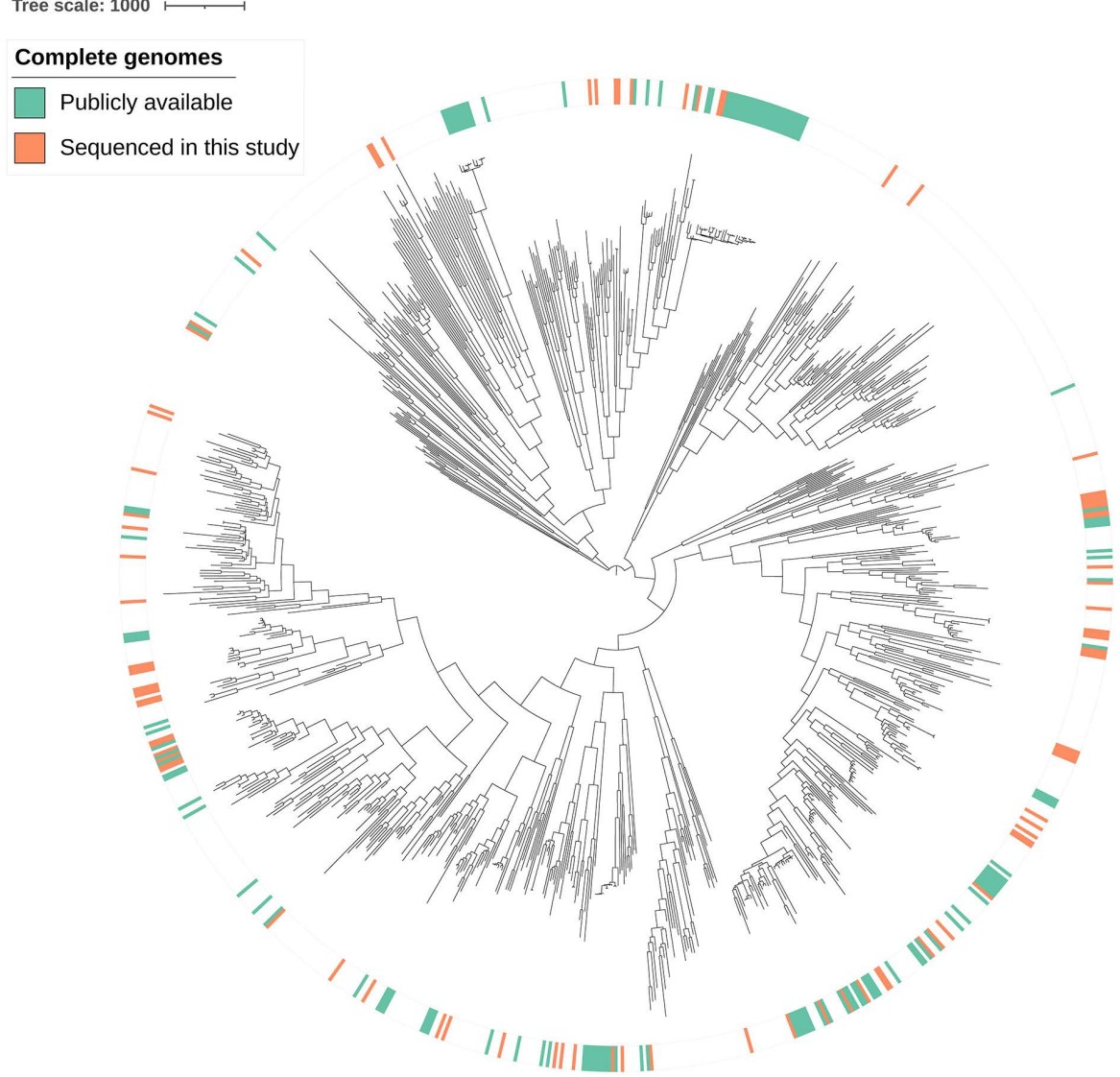

**Tree scale: 1000** ├──────┤

**Complete genomes**
- 🟩 Publicly available
- 🟧 Sequenced in this study

**Fig 1. A diverse set of *N. gonorrhoeae* complete genomes.** Recombination-masked phylogeny of the publicly available complete genomes (generated with long- and short-read sequencing data, green) and the new complete genomes generated in this study (generated with Oxford Nanopore sequencing, orange), in the context of a representative global collection of *N. gonorrhoeae* isolates (generated with short-read sequencing data). The tree scale indicates the number of recombination-masked SNPs.

pangenome clustering of the Prokka annotations. Using BLAST to search for the conserved region of the *opa* gene did not identify any additional intact *opa* genes compared to our method, but it did identify 35 additional *opa* pseudogenes that were missing the beginning or end of the gene and 2 *opa* genes with a mutation in the start codon (ACG instead of ATG). Our method identified 8 *opa* genes not found by BLAST that were divergent from other *opa* sequences. These genes had the pentanucleotide repeats but were divergent in the conserved and semivariable parts of the *opa* sequence. Gene annotations using Prokka identified the same 37 *opa* pseudogenes or *opa* genes with a mutation in the start codon found by the BLAST search, but no additional genes. Our method identified 2 additional *opa* genes that were not annotated by Prokka. Pangenome clustering identified the 37 *opa* pseudogenes or *opa* genes with a mutation in the start codon found

by the BLAST search and one additional *opa* pseudogene. Our method identified 7 *opa* genes not found in the pange-nome clusters, including the 2 *opa* genes that were not annotated by Prokka. In summary, our method of searching for *opa* genes identified the most comprehensive set of *opa* genes from the genomes and, to our knowledge, did not miss any intact *opa* genes.

## Consistent local genomic positioning of *opa* reveals patterns of copy number variation

We next determined the number and locations of *opa* genes in each genome. Out of 219 genomes, 5 genomes (2.3%) had 9 *opa* genes, 41 genomes (18.7%) had 10 *opa* genes, 172 genomes (78.5%) had 11 *opa* genes, and 1 genome (0.5%) had 12 *opa* genes. The genome with 12 *opa* genes (WHO_T_2024) contained a duplication of 47,339 nucleotides with 2 SNPs that led to the exact duplication of one of the *opa* genes. Based on the genomes in this study, we observed *opa* losses up to around 15 times across the phylogeny (Fig 2a). In addition, there were 35 sequences that contained part of an *opa* gene. Of these, 28 occurred in a phylogenetically related cluster of isolates and were missing the last 260

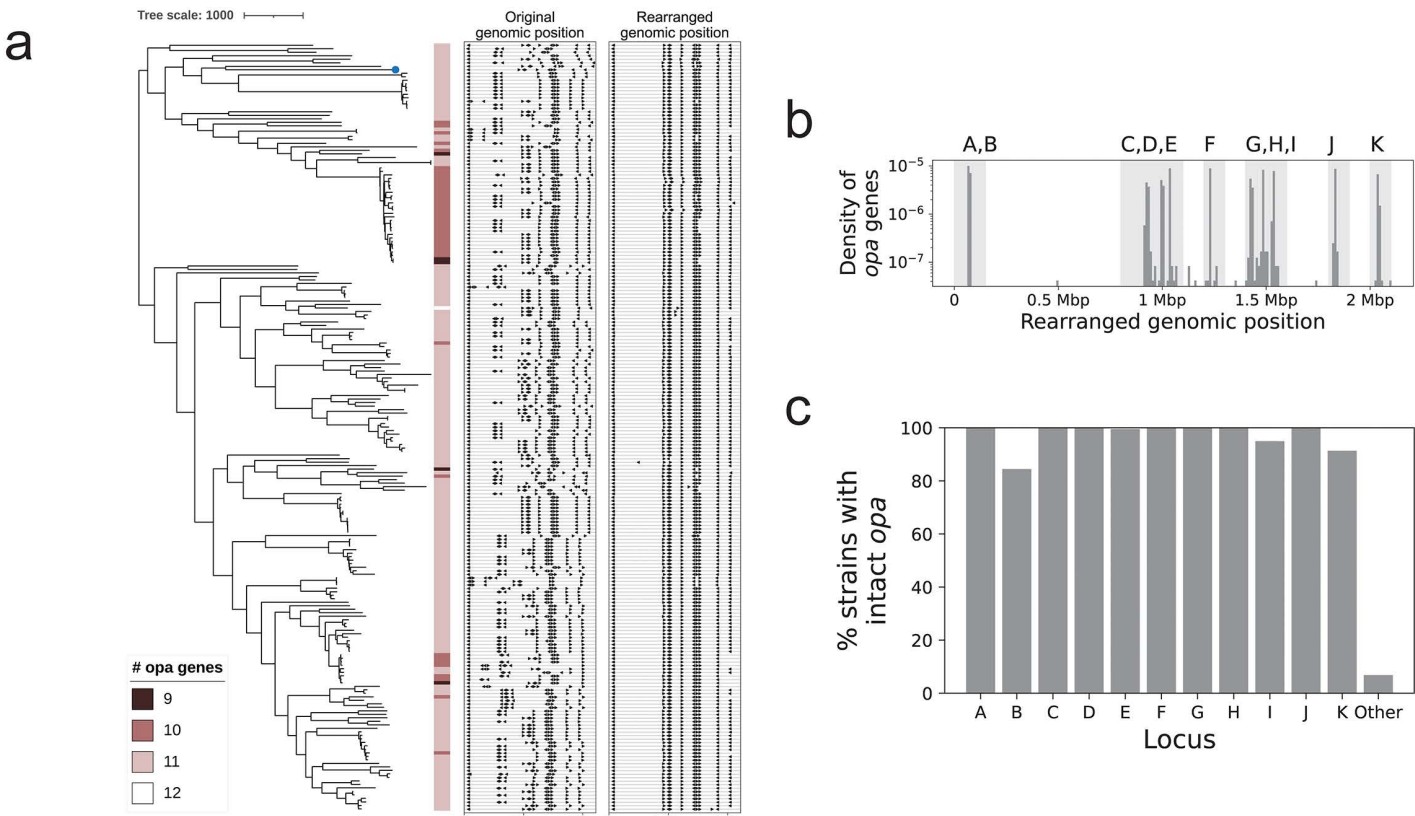

**Fig 2. Consistent local genomic positioning of *opa* reveals patterns of copy number variation. (a)** Recombination-masked phylogeny of complete genomes annotated with the number of intact *opa* genes (contain start codon, stop codon, and N-terminus) in each genome. The genomic locations of the *opa* genes are shown in the middle panel and the rearranged genomic locations of the *opa* genes are shown in the right panel. The reference genome (FA1090) is shown with a blue dot on the phylogeny. The tree-scale indicates the number of recombination-corrected SNPs. **(b)** The distribution of number of *opa* genes by rearranged genomic position. The loci were named *opaA* to *opaK* based on the FA1090 genomic order. *opa* genes that fell within the light-gray shaded background regions were assigned the corresponding name(s). Where multiple *opa* genes occurred in proximity (for exam-ple, A and B; C, D, and E; or G, H, and I), the *opa* genes within each genome were assigned in the order that they appeared in the light-gray shaded region (i.e., the first *opa* gene is C, the second *opa* gene is D, and the third *opa* gene is E). This was done to account for slight shifts in *opa* location across genomes. **(c)** *opa* loci occupancy.

nucleotides of the gene, including the hypervariable 2 region, in the context of an approximately 400 nucleotide deletion that also included 140 nucleotides of sequence downstream of the *opa* gene. These 28 clustered, *opa* pseudogene sequences were similar but not identical and occurred in similar locations in the genome. There were also 2 sequences with a mutation in the start codon (ACG instead of ATG) that otherwise were full length *opa* genes. All but one of the *opa* pseudogene sequences and both *opa* sequences with a mutation in the start codon occurred in genomes that had fewer than 11 intact and full length *opa* genes. If all genomes were to have at least 11 *opa*, then there would be at least 51 missing full length *opa* across the dataset, 37 of which are accounted for here.

The genomic locations of *opa* genes were variable across genomes (S6 Fig), with a mean distance of 59,460 nucleotides between *opa* gene loci across strains. We observed that the pattern of genomic organization of *opa* loci clustered phylogenetically (Fig 2a). To test whether differences in *opa* locations were due to large-scale genomic rearrangements, we used alignments to the reference strain FA1090 to detect and then correct for such rearrangements (S7 Fig). We found 2,331/2,348 (99.3%) *opa* genes within locally collinear alignment blocks with the FA1090 genome (note that the 11 FA1090 *opa* genes have been removed from this calculation). The *opa* genes that did not fall into any locally collinear alignment blocks with the FA1090 genome are listed in S7 Table. The mean distance between *opa* locations in the rearrangement-corrected assemblies decreased to 2,886 nucleotides (Figs 2a and S6). The consistency of *opa* locations in the rearrangement-corrected assemblies allowed us to define 11 loci, which we named in the order that they appeared in FA1090 from *opaA* to *opaK* (Fig 2b). *opa* genes that did not fall within the definitions of the loci were given the label "Other" (Fig 2b and 2c and S8 Table). Loss of full-length *opa* genes occurred most often in *opaB* (16% of all isolates), *opaK* (9% of all isolates), and *opaI* (5% of all isolates) (Fig 2c). There were 15/2,359 (0.6%) *opa* that did not fall within these 11 loci.

Of the 2,359 *opa* genes, 156 (6.6%) had a frameshift mutation after the end of the coding repeat sequence, leading to a stable premature stop codon independent of phase variation. This included at least one *opa* gene in 51/132 (39%) of publicly available genomes and 29/87 (33%) genomes sequenced in this study. None of the superseded WHO reference strains [61] had these frameshift mutations. For the genomes sequenced in this study, there was a negative correlation between the sequencing coverage and the number of *opa* with these frameshift mutations (S8a Fig, Spearman correlation coefficient -0.45, $p = 10^{-5}$), suggesting that some of these frameshift mutations may be due to sequencing error. The premature stop codons were concentrated in three locations in the *opa* gene (S8b Fig) and closer inspection of the nucleotide sequences revealed that variations in the semivariable region sequence upstream of the premature stop codon led to the frameshift mutations.

## Diversity of *opa* alleles within genomes, across genomes, and compared with genome diversity

*opa* genes had almost as much within-isolate diversity as between-isolate diversity, as measured by pairwise amino acid sequence identity (Fig 3a). The number of distinct *opa* allele types per isolate, as defined by having less than 95% amino acid identity, ranged from 3 to 10, with a mean of 7 (Fig 3c). The Shannon diversity of *opa* types in each isolate ranged from 1.03 to 2.27 with a mean of 1.84 (S9 Fig). There were more instances of similar *opa* genes in the same isolate than in different isolates, and 96.8% (212/219) of isolates had at least one pair of *opa* genes with more than 95% amino acid sequence identity (Fig 3b). The redundant *opa* genes were not always found in the same location in the genome and did not have similar sequences to one another (S10 and S11 Figs). One isolate (NG250) had 7 *opa* genes with >95% amino acid sequence identity and three isolates (CT213, TUM15748, and WHO_T_2024) each had 6 *opa* genes with >95% amino acid sequence identity. Thirteen isolates had 4 pairs of *opa* genes with >95% amino acid sequence identity (S12 Fig). *opa* sequence similarity correlated with genome similarity (Spearman correlation coefficient 0.24, p < 0.001) (Fig 3c). In pairs of non-identical isolates, the number of sites in the *opa* genes that differed was 13–3,294x higher (mean: 74x higher) than the number of whole genome sites that differed, showing that *opa* genes are substantially more diverse than the rest of the genome (Fig 3d).

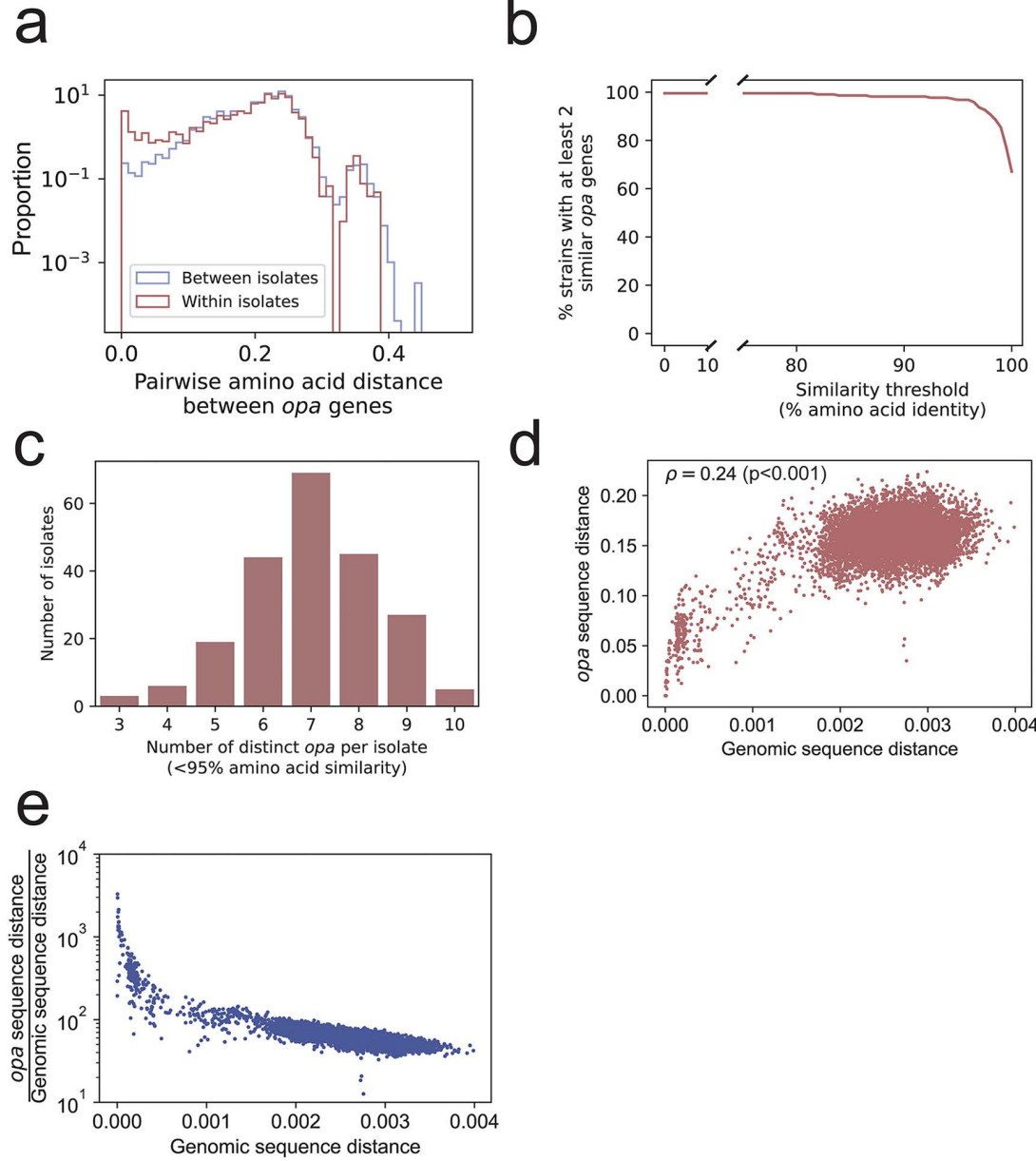

**Fig 3. Diversity of *opa* alleles within genomes, across genomes, and compared with genome diversity. (a)** The distribution of the pairwise amino acid distance between *opa* genes in the same isolate (red) and in different isolates (blue). The distribution of within-isolate *opa* diversity was similar to the distribution of between-isolate *opa* diversity, except for the presence of more near-identical *opa* genes in the same isolate. **(b)** The percentage of isolates with at least 2 *opa* sequences above a given amino acid sequence identity. Nearly all (97.2%) isolates have at least 2 *opa* sequences with >95% amino acid sequence identity. **(c)** The distribution of the number of distinct *opa* allele types per isolate, defined as less than 95% amino acid sequence identity. **(d)** Comparison of *opa* sequence distance and genomic sequence distance in pairs of complete genomes with the same number of *opa* genes. Each point represents one pair of complete genomes with the same number of *opa* genes. A distance of 0 indicates that they are identical, and a distance of 1 indicates every site is different. The Spearman correlation coefficient and p-value are shown in the upper left. **(e)** The same data as in (d) but with the *opa* sequence distance divided by the genomic sequence distance on the y axis.

## Fewer *opa* genes are in frame than expected by chance

193/219 (88%) isolates had at least 1 *opa* gene in frame (Fig 4). The mean number of *opa* genes in frame across all isolates was 1.7 and the maximum of *opa* genes in frame in any isolate was 5. The mean number of in-frame *opa* genes per isolate was significantly lower (p < 0.001, Wilcoxon signed-rank test) than expected from a simple null model where each *opa* has a 1/3 chance of being in frame (mean of 3.4 in-frame *opa* genes per isolate in null distribution). *opa* genes that were more than 95% identical in amino acid sequence to at least one other *opa* in the same isolate were 1.54 times less likely to be in frame than other *opa* genes in the isolate (p < 10^-4) (S13 Fig).

## *N. gonorrhoeae opa* are phylogenetically distinct from *Neisseria* species *opa*

As *N. gonorrhoeae* can gain new genetic diversity through interspecies mosaicism [62,63], we evaluated the possible contribution of other *Neisseria* to *N. gonorrhoeae opa* diversity (S14 Fig). Our approach to searching for the *opa* genes in *Neisseria* species using conserved sequences identified the same number of intact *opa* genes as BLAST, and BLAST identified one additional *opa* pseudogene. Most *N. meningitidis* isolates had 4 full length *opa* (115/136 isolates), and more rarely 3 full length *opa* (19/136 isolates) or 5 full length *opa* (2/136 isolates). *N. flavescens* and *N. lactamica* isolates had 2 full length *opa* (1 isolate of *N. flavescens* and 4 isolates of *N. lactamica*), and *N. polysaccharea* had 1 full length *opa* (1 isolate) (Fig 5a). Multiple *Neisseria* species had partial *opa* genes that contained the C-terminal region of the *opa* gene only: *N. animalis* (2 isolates with 2 partial *opa* each), *N. arctica* (1 isolate with 2 partial *opa*), *N. brasiliensis* (1 isolate with 1 partial *opa*), *N. sicca* (1 isolate with 1 partial *opa*, 2 isolates with no *opa*), *and N. zalophi* (1 isolate with 1 partial *opa*). Additionally, 11/136 *N. meningitidis* isolates had a partial *opa* which all have the same sequence (including the upstream region). *N. gonorrhoeae opa* were phylogenetically distinct from *opa* from other *Neisseria* species, except for one allele, which was found in 8 *N. gonorrhoeae* isolates. This allele clustered with, and its sequence was similar to, commensal *Neisseria opa* (Fig 5b). We note that these 8 *opa* genes were divergent in the conserved parts of the sequence compared to other *N. gonorrhoeae opa* and were the same 8 *opa* genes missed by BLAST in our original *opa* search.

## The rate of *opa* evolution is dependent on the genetic locus

To estimate the rate of evolution, we first clustered *opa* by sequence similarity (S15 Fig). Because of known recombination-mediated shuffling [10,11] of the semivariable, hypervariable 1, and hypervariable 2 sequences

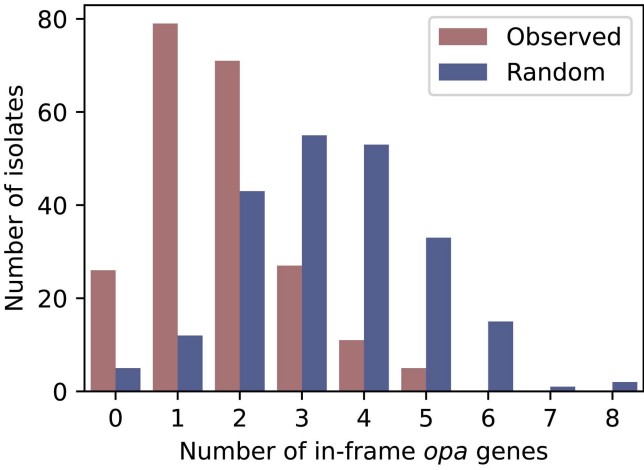

**Fig 4. Fewer *opa* genes are in frame than expected by chance.** The distribution of in-frame *opa* genes per isolate.

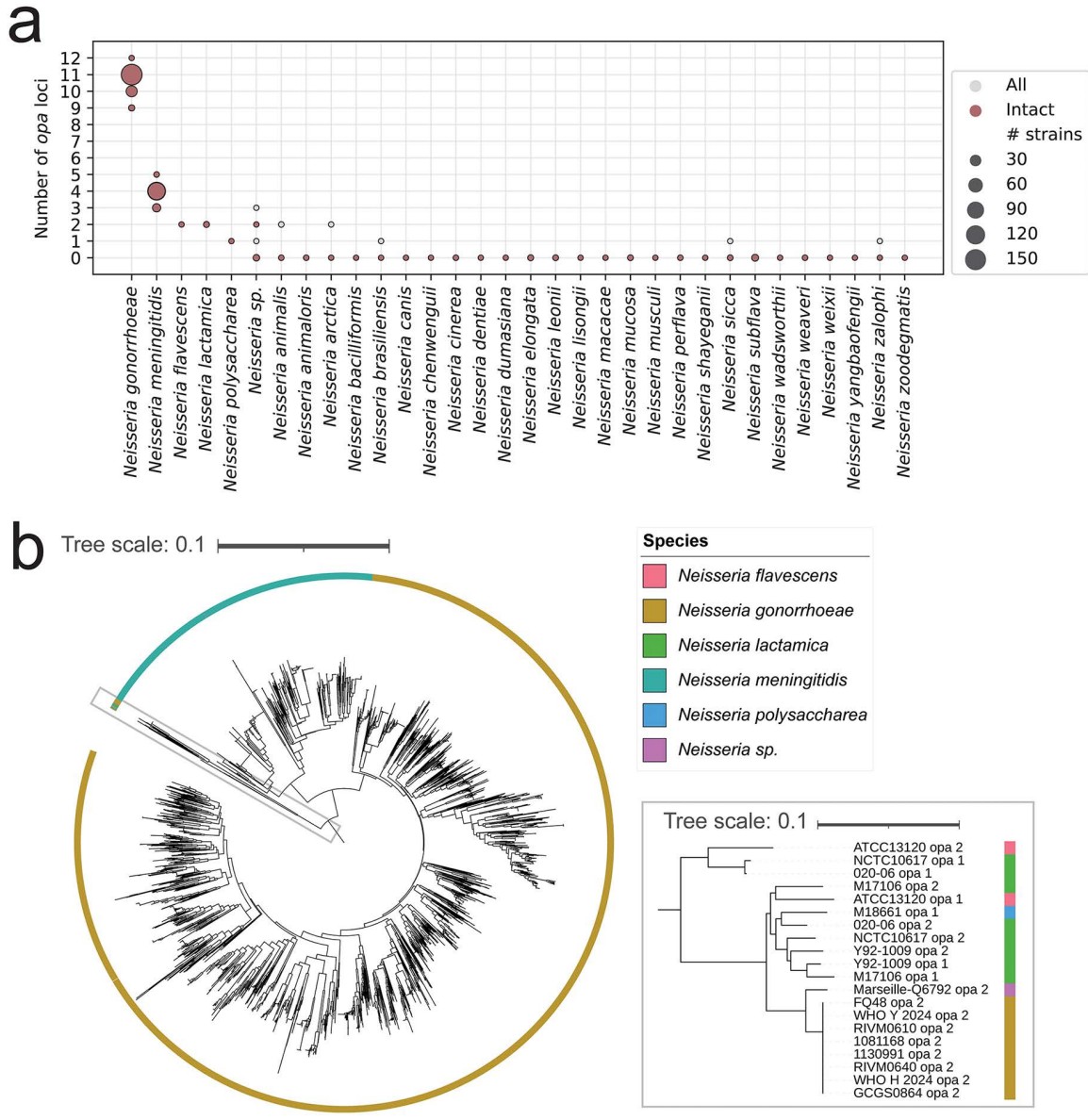

**Fig 5. *N. gonorrhoeae opa* are phylogenetically distinct from other *Neisseria* species *opa*. (a)** The distribution of the total (partial and intact) and intact number of *opa* loci by species. **(b)** k-mer distance-based neighbor-joining phylogeny of the *opa* sequences in all *Neisseria* species. *N. gonorrhoeae opa* are phylogenetically distinct from *opa* sequences from other *Neisseria* species, with the exception of one allele that clustered with other *Neisseria* species *opa* (inset, which shows zoom of the boxed section of the phylogeny). The tree scales indicate the k-mer distance.

(S15a Fig), we clustered each of these regions separately using k-mer distances and a network clustering approach (S15b Fig). The clustering resulted in 6 semivariable region clusters, 21 hypervariable 1 region clusters, and 9 hypervariable 2 region clusters (S15c Fig). The nucleotide k-mer distances for sequences in the same cluster are lower than the distances for sequences in different clusters (S16 Fig), and examination of the sequences shows that each cluster has a distinct motif (S17–S19 Figs).

We reconstructed the ancestral states of the *opa* cluster types on a dated phylogeny to estimate the rate of *opa* evolution. We focused on the only clade in the phylogeny of complete genomes that had densely sampled isolates (branch length less than 200 recombination-corrected mutations and more than 20 samples) (Fig 6a). The inferred substitution rate, $3.1 \times 10^{-6}$ [$1.7 \times 10^{-6}$, $4.4 \times 10^{-6}$] substitutions per chromosome per year, fell within the rate expected for *N. gonorrhoeae* [64]. The rate of cluster type changes was substantially higher for *opaK* compared to the other *opa* loci (Fig 6b). The cluster types that appeared in *opaA* and *opaK* were similar, showing that the difference in the rates of cluster type changes between the two loci was due to different rates of shuffling between the same cluster types (S20 Fig). *opaK* was not significantly more in frame than the other *opa* loci (S21 Fig).

### Correlation of hypervariable 1 and hypervariable 2 region alleles

Using the cluster typing, we were also able to assess whether certain hypervariable 1 and hypervariable 2 alleles appeared together more often than expected by chance (S22 Fig). We accounted for isolate sampling and population structure by randomly sampling 1 isolate per whole genome cluster 100 times. For each subset of representative isolates, out of 100 randomizations of the data, the actual data always showed higher levels of association between hypervariable types than the randomized data.

## Discussion

### Quantitative assessment of *opa* diversity, reading frame, and evolution

Prior to this study, there were several open questions about *opa* genes in *N. gonorrhoeae*: Are there patterns in *opa* diversity and reading frame? How are *opa* genes organized in the genome? How much *opa* diversity is there in the population? How do *opa* evolve? Here, we have advanced our understanding for each of these questions using a new diverse dataset of complete *N. gonorrhoeae* genomes.

**Number, phase, and locations of *opa* genes.** While most isolates had 11 *opa* genes, we found a handful of instances of *opa* loss and gain. The loss events occurred across the phylogeny. In some instances, we saw examples of the gene loss events passed on to progeny. More than 50% of these loss events were explained by *opa* pseudogenes that were the beginning or end of the gene. We were unable to identify the mechanism for the other loss events, which may have resulted from larger deletions in surrounding regions of the genome, since we could not find remnants of these *opa*

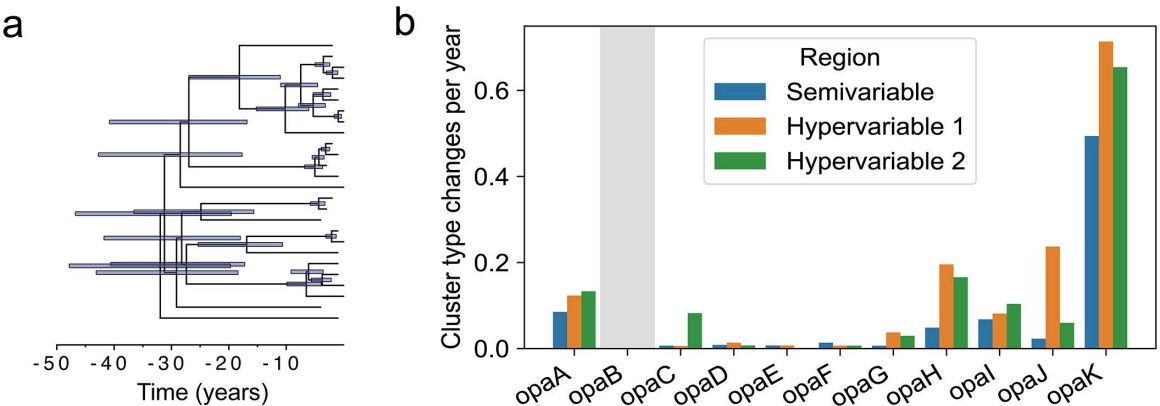

**Fig 6. The rate of *opa* evolution is dependent on the genetic locus. (a)** Dated phylogeny of the clade in Fig 2a with densely sampled isolates (branch length less than 200 recombination-corrected mutations and more than 20 samples). Error bars represent the 95% highest posterior density of the node heights. **(b)** Rate of semivariable, hypervariable 1, and hypervariable 2 cluster type changes by *opa* locus for subtree depicted in (a). The gray shaded region denotes that *opaB* was missing from all isolates in this subtree. In other loci, there were no more than 2 isolates missing each *opa*.

genes by homology search methods. Furthermore, in intact *opa* sequences, we observed frameshift mutations occurring downstream of the coding repeat sequence leading to a premature stop codon, but we were unable to determine how many of these frameshift mutations were due to sequencing error. Our simulation studies indicated that genome coverages down to 50x are reliable and those down to 21x could also be reliable. The genomes that we sequenced have coverage above this range. We were unable to conclusively determine the coverages of the publicly available complete genomes. While our simulation tests suggest that frameshift mutations due to sequencing error at the coverage levels for the complete genomes that we sequenced should be rare, we could not fully resolve the issue for the observed frameshift mutations. In one isolate, we identified 12 *opa* genes, which seemed to be due to a recent genomic duplication event that exactly duplicated another *opa* gene in the genome. The fact that we only observed one isolate with 12 *opa* suggests that the expansion of *opa* number is rare.

Across the population, fewer *opa* were in frame than expected by chance. Furthermore, near-identical *opa* genes in the same isolate tended to be out of frame more often than unique *opa* genes, suggesting selection against expression of redundant *opa*; we note, however, that the receptor binding characteristics of Opa proteins cannot be predicted from *opa* sequence alone and there may be additional phenotypic redundancy, i.e., Opa proteins with low sequence homology may also share receptor binding characteristics. We further note that the isolates we sequenced were not specifically selected for Opa expression state during culturing, so the average number of in frame *opa* genes may be lower than observed in isolates directly obtained from human specimens with no or minimal passaging. Similarly, publicly available genomes may have an over representation of out of frame *opa* genes due to unknown culturing conditions. Nevertheless, our results suggest mechanisms other than chance alone guide the patterns of phase variation, for example, regulation of changes in repeat number, selection against isolates with many *opa* alleles in frame, or selection for Opa proteins with specific receptor binding characteristics. While these mechanisms are yet unknown, previous work suggested that promoter strength may play a role in regulating *opa* phase [26], and this could potentially regulate changes in repeat number. Another potential mechanism is that there may be selection against in-frame *opa* in certain environments, for example the presence of progesterone [65–67], which has been shown to inhibit the growth of Opa positive isolates *in vitro*.

Genomic rearrangements shuffle *opa* position in the genome, but by accounting for them, we were able to assign consistent *opa* loci. Genomic rearrangements in closely related isolates happened very rarely, but the locus assignment approach can be updated if we detect more examples of genomic rearrangements in closely related isolates.

***opa* diversity and evolution.** The diversity of *opa* alleles within isolates was generally high, with the average isolate having 7 distinct *opa* alleles, but there were some isolates with as few as 3 distinct *opa* alleles. This suggests that there is some pressure to have diverse *opa* alleles within an isolate. At the same time, almost all isolates had at least two near-identical *opa* alleles, possibly as a result of gene conversion, suggesting there may be a balance between maintaining diversity and maintaining function while diversifying.

*opa* genes are on average 74 times more diverse than the rest of the genome. More closely-related isolates have more similar *opa* sequence repertoires. In the context of reinfections, our results suggest that when reinfections occur with the same or a closely related isolate, there will be a similar *opa* repertoire. We speculate that if there is immunological memory to the *opa* that are expressed, then there will be selection in subsequent reinfections for isolates that have different *opa* turned on, an idea that is also supported by evidence of variation in *opa* expression over the course of single infections in human challenge studies [68,69]. When reinfections occur with different isolates, there will likely be different *opa* alleles expressed.

*N. gonorrhoeae opa* allelic diversity did not overlap with *opa* from other *Neisseria* species. While sequencing more complete genomes may lead to more observations of interspecies mosaicism, the separate clustering of *N. gonorrhoeae* and *N. meningitidis opa* in our dataset and in Malorny et al. [9] suggests phylogenetic separation of *opa* sequences between species. It is possible that interspecies recombination was historically important in shaping the *opa* repertoire and that the phylogenetic separation of *opa* between species may reflect ancient divergence. The separate clustering of

*Neisseria* species *opa* may be due to the different functional constraints of host receptor binding or differences in flanking regions of *opa* genes from different species that prevent homologous recombination from occurring between species.

*opaK* varies more rapidly than other loci, but it is not significantly more in frame in our dataset. The differences in rates of evolution by *opa* locus may be due to variability in the accessibility of each locus for recombination or other diversity generating processes (i.e., structural accessibility). The changes in *opaK* tend to reflect recombination among existing *opa* types, suggesting that diversity is mostly generated through shuffling of existing sequences. Bilek et al. [24] suggested a possible connection between *opa11* (*opaK* in this study) variation and the neighboring *pilE* Pilin expression gene variation, but noted this may not be causal.

Hypervariable region 1 and 2 alleles in *N. meningitidis* are more correlated than expected due to chance [70]. We showed that this phenomenon also occurs in *N. gonorrhoeae*, which is consistent with studies showing non-random associations of particular hypervariable region 1 and 2 sequences [71,72]. The non-random association of hypervariable alleles is consistent with predictions of a mathematical model of strong immune selection [73], physical linkage on the chromosome, and selection for particular combinations of hypervariable alleles for functional reasons (i.e., to bind to specific human receptors) [74]. Previous work has shown that swapping the hypervariable region sequences between *opa* genes does not lead to predictable changes in receptor binding [74], underscoring the importance of functional coordination between these regions and providing a plausible mechanism for the selection of specific combinations.

### Limitations of our study

There are several limitations to our study. First, the number of complete genomes sequenced was limited, possibly missing some *opa* diversity. Second, gonococcal passaging during culturing may have changed the phase of *opa* genes and specifically could have decreased the number of in frame *opa* genes due to Opa not being selected for in laboratory culture. Culture-independent sequencing that resolves *opa* phase *in vivo* will add to our understanding of patterns of *opa* expression. Third, our study is observational and we have not performed experiments to test the model shown in Fig 7. Fourth, there is limited data on *opa* function, with only a few *opa* sequences characterized for their binding to human receptors and antibody recognition [8,11,15]. While the clustering nomenclature for *opa* that we developed is entirely sequence based, we expect closely related sequences to have more closely related functions on average. However, this classification scheme may miss some phenotypic redundancy, as Opa proteins that lack high sequence homology can retain receptor binding characteristics. Currently, the receptor binding properties of Opa proteins cannot be predicted from *opa* sequence. In the future, functional data can be incorporated to refine the *opa* clustering definitions. Fifth, we were

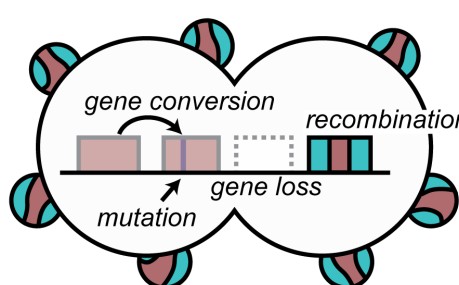

**Fig 7. Model of *opa* variation and evolution supported by our results. (a)** On short timescales, phase variation modulates the frame of highly diverse *opa* alleles within isolates, typically with very few *opa* in frame at one time. **(b)** On long timescales, recombination, gene conversion, mutation, and gene gain and loss within species in a locus-dependent manner lead to changes in the *opa* repertoire.

limited in the estimation of the rate of evolution to only a single densely sampled subtree. There may be differences in the rate of *opa* evolution across genomic backgrounds.

### Model

Taken together, our results support a model of *opa* variation where on short timescales phase variation modulates the translation of highly diverse *opa* alleles, typically with very few *opa* in frame at one time. On long timescales, recombination, gene conversion, mutation, and gene gain and loss lead to changes in the *opa* repertoire, with the rate of these processes dependent on the genetic locus (Fig 7). The rapid rate of phase variation and diversification will likely lead hosts to be exposed to different Opa types upon reinfection and may allow antigenic escape *in vivo* in chronic or undiagnosed infections.

## Supporting information

**S1 File. Supplementary methods and results.**
(DOCX)

**S1 Fig. Flow chart summarizing algorithm for identifying *opa* genes.** Note that a mismatch refers to a substitution, insertion, or deletion.
(PNG)

**S2 Fig. The distribution of continents, host genders, and years of isolation for the isolates that we sequenced from across the *N. gonorrhoeae* phylogeny (blue) and the publicly available complete genomes (orange).** The information on host gender was not included with NCBI metadata for the publicly available complete genomes. Abbreviations: Man (M), Woman (W), Not available (NA).
(PDF)

**S3 Fig. Comparison of *opa* sequences across long-read assembly and polishing methods.** The number of changes in *opa* genes at 4 read coverage levels using 7 different assembly and polishing procedures. There are two points of the same color in each coverage level, indicating two different isolates. The changes at 21x for methods 1 and 2 include one SNP in one isolate's genome and one single base insertion in the other isolate's genome.
(PDF)

**S4 Fig. More extensive comparison of *opa* sequences in Autocycler assemblies across 11 phylogenetically diverse isolates at 4 read coverage levels.** The number of changes in *opa* genes at 4 read coverage levels using Autocycler. For each isolate, the reads were randomly subsampled 10 times at each read coverage (replicates) and an assembly was created with the subsampled reads using Autocycler. The size of the point indicates the number of replicates. The changes at 50x coverage include one SNP in two separate genomes and one undetected *opa*, the change at 75x was multiple sequence differences in one genome, and the change at 125x coverage was one SNP in one genome.
(PDF)

**S5 Fig. The distribution of read coverages for the genomes that we sequenced in this study.**
(PDF)

**S6 Fig. The distance between *opa* genes is higher in the original positions than after accounting for genomic rearrangements.** The distances are calculated as the distance between each *opa* in the reference genome FA1090 to the *opa* that is closest in genomic position in all other isolates that had 11 *opa*.
(PDF)

**S7 Fig. Schematic of genomic rearrangement procedure (see Methods).** The shaded colored regions are the locally collinear blocks (LCBs). The LCBs that appear above the gray line are on the forward strand and those that appear below the gray line are on the reverse strand. A triangle pointing to the right indicates an *opa* gene on the forward strand and a triangle pointing to the left indicates an *opa* gene on the reverse strand.
(PDF)

**S8 Fig. A subset of *opa* genes exhibit frameshift mutations after coding repeats leading to a premature stop codon.** (a) The approximate genome sequencing coverage and number of *opa* in the genomes with frameshift mutations downstream of the coding repeats. (b) The locations of the stop codons in the *opa* genes with frameshift mutations downstream of the coding repeats. The maximum value of the x-axis is set at the average length of *opa* amino acid sequences.
(PDF)

**S9 Fig. Shannon diversity of *opa* types within isolates.**
(PDF)

**S10 Fig. Redundant *opa* genes are not always found in the same locations across isolates.** The rearranged genomic locations (using FA1090 as the reference genome) of the *opa* genes that are redundant (>95% amino acid identity within the same isolate) and *opa* genes that are unique (≤95% amino acid identity within the same isolate). The unique and redundant *opa* have a similar distribution of genomic locations.
(PDF)

**S11 Fig. Redundant *opa* do not have the same sequences across isolates.** The distribution of pairwise amino acid distances between *opa* genes that are redundant (>95% amino acid identity within the same isolate) and *opa* genes that are unique (≤95% amino acid identity within the same isolate). The pairwise distance was calculated using only pairs of *opa* across isolates to not bias the calculation for redundant *opa*, which by definition have low pairwise distances when comparing within isolates. The distributions of pairwise distances are similar for unique and redundant *opa*, implying that the redundant *opa* do not have the same sequences across isolates. The peak at the right of the unique *opa* distribution is due to the 8 highly divergent *opa* sequences which do not have a redundant copy in the same isolate.
(PDF)

**S12 Fig. Genomes exhibit distinct patterns of similar *opa*s.** All isolates with complete genomes are shown on the x-axis. The points indicate groups of *opa* genes in the same isolate with >95% amino acid sequence identity. The y-axis shows the number of similar *opa* genes in each group. The size and color of the point indicate the number of distinct groups of each size in the genome. The x-axis is sorted first by the maximum number of similar *opa* genes in any group and then by the maximum number of groups. The plot is split into three rows for readability.
(PDF)

**S13 Fig. The *opa* that are unique in an isolate are more likely to be in frame than the *opa* that are redundant.** (a) The percentage of *opa* that are in frame for *opa* that are unique within an isolate (<95% amino acid similarity) or redundant within an isolate (≥95% amino acid similarity). (b) The ratio of the percentage of unique *opa* in frame to the percentage of redundant *opa* in frame for $10^4$ randomizations of the data. The randomization procedure permuted which *opa* are labeled redundant, keeping the same total number of redundant *opa* across all isolates. Zero randomizations gave ratios as high as in the actual data (ratio of 1.54, indicated by the vertical dashed black line) yielding a p-value of less than $10^{-4}$.
(PDF)

**S14 Fig. The number of publicly available complete genomes by species from the *Neisseria* genus.**
(PDF)

**S15 Fig. Network-based clustering approach for semivariable and hypervariable *opa* sequences.** (a) Schematic of the *opa* gene. The exact length and locations of the gene features varies across *opa* genes; depicted here is FA1090 *opa1*. CR: coding repeat, SV: semivariable region, HV1: hypervariable 1 region, HV2: hypervariable 2 region. (b) Summary of the approach to clustering variable region sequences. For each variable region (semivariable, hypervariable 1, and hypervariable 2), we calculated the k-mer distances between all sequences using MASH, setting k such that the probability of finding a random k-mer in each sequence is 0.01. We performed successive rounds of inflation (expansion and contraction) on the distance matrix, which amplifies high values of the matrix and suppresses low values of the matrix. We chose the lowest inflation parameter that gave a stable clustering. (c) The distribution of the cluster types for the sequences in each variable region.
(PDF)

**S16 Fig. Sequences generally are more similar within clusters than between clusters.** The mean nucleotide distance between pairs of sequences in the same and different clusters in the semivariable (a), hypervariable 1 (b), and hypervariable 2 (c) regions.
(PDF)

**S17 Fig. The sequence logos for each cluster of the semivariable sequences.** The nucleotide sequences were aligned using MAFFT in each cluster. The height of the nucleotides represents the number of sequences with each nucleotide.
(PDF)

**S18 Fig. The sequence logos for each cluster of the hypervariable 1 sequences.** The nucleotide sequences were aligned using MAFFT in each cluster. The height of the nucleotides represents the number of sequences with each nucleotide.
(PDF)

**S19 Fig. The sequence logos for each cluster of the hypervariable 2 sequences.** The nucleotide sequences were aligned using MAFFT in each cluster. The height of the nucleotides represents the number of sequences with each nucleotide.
(PDF)

**S20 Fig. *opaA* and *opaK* have similar cluster types that change at different rates.** The dated phylogeny of the subtree shown in Fig 6b annotated with the cluster types for the semivariable (a-b), hypervariable 1 (c-d), and hypervariable 2 (e-f) regions shown as colored circles for opaA (a, c, e) and opaK (b, d, f). The colors are comparable within each sequence region across loci (e.g., the same color scheme is used for the semivariable region of both *opaA* and *opaK*) but not between sequence regions. The colored circle annotations at the tips represent the cluster types of the isolates, and the colored circle annotations in the internal nodes represent the inferred ancestral state by PastML. The cluster types that are present in *opaA* and *opaK* are similar, but more closely related isolates have more similar cluster types in *opaA* compared to *opaK*.
(PDF)

**S21 Fig. No significant differences in fraction of *opa* in frame by locus.** One-sided proportions Z-tests with a Bonferroni multiple hypothesis correction comparing the fraction of *opa* in frame at each locus (1) to the total fraction of *opa* in frame across all loci and (2) to the fraction of *opaK* alleles that are in frame are not significant.
(PDF)

**S22 Fig. Non-random association of hypervariable 1 and hypervariable 2 allele types in *opa* genes, as shown for one representative random subset of 1 isolate per BAPS cluster.** (a) The number of each combination of

hypervariable 1 and hypervariable 2 types in *opa* alleles after accounting for isolate sampling and population structure. The columns of the matrix were rearranged to give the largest matrix trace (padding with columns of all zeros to make a square matrix). (b) The same data representation in (a) but after randomizing the assignment of hypervariable 2 types across the opa alleles. (c) The distribution of the maximum sum of the diagonal matrix elements (allowing for column rearrangements) in 100 randomizations of the hypervariable 2 type (blue histogram) compared to the actual data (black dashed line).
(PDF)

**S1 Table. Publicly available complete genomes.**
(XLSX)

**S2 Table. Complete genomes sequenced in this study.**
(XLSX)

**S3 Table. References for publicly available *N. gonorrhoeae* short-read sequencing data meeting quality control thresholds for selection of representative draft genomes.**
(XLSX)

**S4 Table. Representative *N. gonorrhoeae* draft genomes.**
(XLSX)

**S5 Table. Publicly available *Neisseria* species complete genomes.**
(XLSX)

**S6 Table. The NG-STAR types, associated antibiotic resistance markers, and available antibiotic minimum inhibitory concentrations for the genomes used in this study.**
(XLSX)

**S7 Table. *opa* genes that did not fall into any locally collinear alignment blocks with the FA1090 genome.**
(XLSX)

**S8 Table. *opa* genes that did not fall within the definitions of the loci and were given the locus label "Other".**
(XLSX)

## Acknowledgments

We are grateful to Eric Neubauer Vickers for helpful discussions and comments on the manuscript. We are grateful to the Grad lab and the Center for Communicable Disease Dynamics for helpful discussions. The computations in this paper were run on the FASRC Cannon cluster supported by the FAS Division of Science Research Computing Group at Harvard University.

## Author contributions

**Conceptualization:** QinQin Yu, Tatum D. Mortimer, Yonatan H. Grad.

**Data curation:** QinQin Yu, Tatum D. Mortimer.

**Formal analysis:** QinQin Yu.

**Funding acquisition:** QinQin Yu, Yonatan H. Grad.

**Investigation:** QinQin Yu, Sofia O.P. Blomqvist, Bailey Bowcutt, Samantha G. Palace.

**Methodology:** QinQin Yu, Tatum D. Mortimer, David Helekal.

**Project administration:** Yonatan H. Grad.

**Resources:** QinQin Yu, Yonatan H. Grad.

**Software:** QinQin Yu, Tatum D. Mortimer, Sofia O.P. Blomqvist, David Helekal.

**Supervision:** Yonatan H. Grad.

**Validation:** QinQin Yu, Sofia O.P. Blomqvist.

**Visualization:** QinQin Yu, Tatum D. Mortimer, Yonatan H. Grad.

**Writing – original draft:** QinQin Yu, Yonatan H. Grad.

**Writing – review & editing:** QinQin Yu, Tatum D. Mortimer, Sofia O.P. Blomqvist, Bailey Bowcutt, David Helekal, Samantha G. Palace, Yonatan H. Grad.

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
