## [Decision Letter · Decision Letter 0]

3 Mar 2026

PPATHOGENS-D-26-00248

Diversity and evolution of a phase-variable multi-locus antigen in Neisseria gonorrhoeae

PLOS Pathogens

Dear Dr. Grad,

Thank you for submitting your manuscript to PLOS Pathogens. After careful consideration, we feel that it has merit but does not fully meet PLOS Pathogens's publication criteria as it currently stands. Therefore, we invite you to submit a revised version of the manuscript that addresses the points raised during the review process.

We look forward to receiving your revised manuscript.

Kind regards,

Mark Davies, Ph.D

Academic Editor

PLOS Pathogens

Debra Bessen

Section Editor

PLOS Pathogens

Sumita Bhaduri-McIntosh

Editor-in-Chief

PLOS Pathogens

orcid.org/0000-0003-2946-9497

Michael Malim

Editor-in-Chief

PLOS Pathogens

orcid.org/0000-0002-7699-2064

**Additional Editor Comments:**

The reviewers acknowledge the sound body of work that this work represents. Please address all comments in your revision.

**Journal Requirements:**

1) Please provide an Author Summary. This should appear in your manuscript between the Abstract (if applicable) and the Introduction, and should be 150-200 words long. The aim should be to make your findings accessible to a wide audience that includes both scientists and non-scientists. Sample summaries can be found on our website under Submission Guidelines:

https://journals.plos.org/plospathogens/s/submission-guidelines#loc-parts-of-a-submission

**Reviewers' Comments:**

Reviewer's Responses to Questions

**Part I - Summary**

Reviewer #1: The authors describe their work on various aspects of the opa genes in gonococci. Containing several opa genes that are subject to antigenic variation due to slip-strand mispairing evolutionary work has been hampered by sequencing technologies that heretofore were unsuitable to address the questions asked by the authors. Basic understanding of the evolution and variation of opa genes is critical for advancing knowledge regarding the contribution made by different Opa outer membrane proteins during infection and different forms of disease. To their credit the sequencing/bioinformatic approaches employed provide new insights. In general, the execution of the work was rigorous and the results are solid. While the scholarship is high I do have a few concerns that require attention.

1. The abstract refers to diverse clinical isolates but there is no description of them with respect to geographical location, gender or age of the infected, year of isolation or antibiotic resistance property. Such information should be provided to give the reader an idea as to the diverse nature of the isolates.

2. The model shown in Fig. 7 is important. Unfortunately, there was seemingly no attempt to perform an in lab evolution experiment to test it. The absence of such a test should be incorporated into the limitations section within the Discussion.

3. While I agree that an Opa protein is likely to be only produced by a phase on gene it would be good to actually show this by western blotting. I suggest this because a promoter mutation could result in down-regulation of the cognate in-frame gene an this could influence biological events during infection.

4. The group recently published a manuscript in Nature Microbiology regarding fitness of gonococci during infection when the infecting strains differ in antibiotic resistance markers. If such recovered gonococci are available the manuscript would be strengthened if they followed the emergence of opa variants during experimental infection in the female mouse model that was used.

Reviewer #2: This is a well-executed study that makes important contributions to our understanding of Neisseria gonorrhoeae opa diversity and evolution. The use of long-read sequencing to resolve multiple opa loci is methodologically sound, and the resulting dataset and analyses are comprehensive. The manuscript is generally well-written, and the findings are significant for vaccine development and immunological studies.

Reviewer #3: This manuscript is a comprehensive investigation of the opacity-associated (Opa) protein gene family of Neisseria gonorrhoeae. Opa proteins are major adhesins and invasins for N. gonorrhoeae on human cells. Using publicly available genomes and genomes from strains sequenced by the authors, the authors develop a robust genomic and phylogenetic pipeline to identify putative opa genes, examine their synteny in aligned genomes, and predict their likelihood of expression based on the gene being in- or out-of-frame. This is an extensive, rich dataset that is a valuable addition to the Neisseria field. The findings, which are supported by the data, are interesting and insightful. I am very enthusiastic about this manuscript. My comments are intended to provide additional clarity and biological context for the general microbial pathogenesis audience of this study.

**Part II – Major Issues: Key Experiments Required for Acceptance**

Reviewer #1: Points 3 and 4: the performance of these experiments is not critical but are only suggested to strengthen an already excellent paper.

Reviewer #2: (No Response)

Reviewer #3: 1. Line 16, short title: Please do not use the term “antigenic variation” when describing Opa phase variation. In Neisseria, antigenic variation is shorthand for the recombination-based changes to the major pilin-encoding gene, so a reader could be confused by using antigenic variation to describe the Opa family. One suggestion is “(Landscape of) Opa variation in Neisseria gonorrhoeae”.

2. The study is inferring expression of Opa proteins from the putative translation of the DNA sequence in the sequenced genomes. However, Opa protein expression itself is not tested. Therefore, places that mention Opa expression should be reworded to “in frame”. This includes lines 50, 635, 637, 641.

3. For Figure 1, strains that were newly sequenced, how were the bacteria grown for isolating genomic DNA? Was there any phenotypic selection for Opa expression state? In vitro passaged bacteria may not be selected for their Opa expression, so the average number of phase-ON opa genes may be lower than observed in isolates directly obtained from human specimens with no or minimal passage. This point should also be addressed in lines 513-516, since publicly available genome sequences may have an over-representation of phase-OFF opa genes.

4. Discussion: With the many subheadings, this section is disjointed, and in some places reiterates the Results without additional context. The Discussion could be reorganized to combine some of the small sections into paragraphs, each with conclusions and future directions associated with them.

a. Line 618, can you speculate the mechanism for the other loss events – are there repeat sequences in or near the opa genes where the losses are seen?

b. Line 628-633: are the duplicated opa genes always found in the same locations in each genome? Is it always the same pairs of opa genes that are duplicated?

c. Line 635 section, and also sentence on line 694-695: Missing is the concept of phenotypic redundancy: that Opa proteins lack sequence homology but retain receptor binding characteristics. In this context, it is important to include that receptor binding property cannot be predicted from opa sequence.

d. Line 652: the terminology implies that N. gonorrhoeae can regulate the on/off expression of opa genes (“turned on in subsequent reinfections”). Opa genes undergo phase variation at random, and then there is selection for or against individual bacteria in a population. This diversity helps the population as a whole to maintain infectivity and survival.

e. Section starting on line 679, add citations for studies describing the non-random association of particular hypervariable loop 1 and 2 sequences (PMID: 1809845, PMID: 1694004). Also add that swapping hypervariable loops does not lead to predictable interpretations of receptor binding (reference 67). And line 693, add references #8 and 11 along with #15.

f. “Genomic rearrangements shuffle opa position” is a less major conclusion than others in this manuscript, and I would not lead off the discussion with this point.

**Part III – Minor Issues: Editorial and Data Presentation Modifications**

Reviewer #1: Please see points 1 & 2 above.

Reviewer #2: Minor Comments

I recommend publication, although the following minor issues should be addressed.

1. The algorithm for identifying opa genes (Supplementary Methods, lines 1079–1099) is complex. Consider adding a flowchart or decision tree in the main Methods section to improve accessibility.

2. The negative correlation between sequencing coverage and frameshift mutations suggests potential sequencing artefacts. The authors acknowledge but do not fully resolve this issue. The authors could provide a more detailed discussion of the threshold coverage at which frameshift mutations become reliable. Consider whether genomes below a certain coverage threshold should be excluded from downstream analyses, or clearly flag them as potentially unreliable.

3. The authors identify 98.8% of opa genes within locally collinear blocks with FA1090, but the remaining 1.2% (28 genes) are not accounted for in the locus assignment scheme. Could the authors clarify how these 28 genes are handled in downstream analyses. Provide a supplementary table listing these outliers and their genomic contexts.

4. The finding that fewer opa genes are in frame than expected by chance (mean 1.7 vs. expected 3.4, p < 0.001) is intriguing but the mechanistic explanation remains speculative. Expand the discussion of potential mechanisms (lines 639–644). The mention of promoter strength and progesterone-mediated selection is interesting but underdeveloped. Consider whether phase variation dynamics (i.e. the rate of frameshift/in-frame transitions) might differ between loci, which could explain the observed pattern.

5. The conclusion that "interspecies recombination does not appear to play a large role" (line 657) is based on the observation that only one N. gonorrhoeae opa allele clusters with commensal Neisseria opa. Acknowledge that this conclusion is based on a single observation and may not be definitive. Discuss the possibility that interspecies recombination events may be rare but historically important in shaping the opa repertoire. Consider whether the phylogenetic separation of opa between species could reflect ancient divergence rather than absence of recombination.

Suggested wording changes:

The manuscript is generally well-written, but a few sentences could be tightened.

6. Line 29: "Fewer opa genes were in frame, and thus inferred to be expressed, than expected due to chance." Consider: "Fewer opa genes were in frame (and thus inferred to be expressed) than expected by chance."

7. Line 72: "Studies of opa variation and evolution have focused on small numbers of isolates or closely related isolates." Consider: "Previous studies of opa variation and evolution have been limited to small numbers of isolates, often closely related."

Reviewer #3: 1. Terminology on Line 21, Line 63: The full name for Opa is opacity-associated protein. Opa proteins are a family of related proteins, so would reword to (line 21) “One of the most abundant and diverse antigens are members of the opacity-associated (Opa) family, surface proteins that mediate gonococcal attachment…” and (line 63) “A major family of surface-exposed proteins that undergo this variation are the opacity-associated (Opa) proteins”.

2. Line 26, Line 360, Line 375, elsewhere: Please define what is “diverse” about the clinical isolates.

3. Line 64: Replace “host receptors” with “human receptors” to emphasize these effects are human-specific. Also add that Opa proteins mediate non-opsonic phagocytosis by human neutrophils (cite PMID: 8971715, PMID: 8962144, PMID: 9218786).

4. Line 68: Add that the pentanucleotide repeats are in the signal sequence-encoding region of the opa gene.

5. Line 77: add the connection between Opa phase variation and opa gene promoter strength, reference 59.

6. Lines 163-165: please clarify this wording.

7. Line 168: Seems like it would be more accurate to call the opa genes missing the start or stop codon pseudogenes, not “truncated,” which implies a premature stop codon or an internal deletion.

8. Line 389: describe what makes these 8 opa genes divergent. Do they have the pentanucleotide repeats and conserved parts of the opa sequence?

9. Line 416: how many is a “handful of times”?

10. Figure on line 485 should be deleted.

PLOS authors have the option to publish the peer review history of their article (what does this mean?). If published, this will include your full peer review and any attached files.

**Do you want your identity to be public for this peer review?** For information about this choice, including consent withdrawal, please see our Privacy Policy.

Reviewer #1: No

Reviewer #2: No

Reviewer #3: **Yes:** Alison Criss

**Figure resubmission:**
---

## [Editor Report · Decision Letter 1]

28 Apr 2026

Dear Dr Grad,

We are pleased to inform you that your manuscript 'Diversity and evolution of a phase-variable multi-locus antigen in Neisseria gonorrhoeae' has been provisionally accepted for publication in PLOS Pathogens.

Best regards,

Mark Davies, Ph.D

Academic Editor

PLOS Pathogens

Debra Bessen

Section Editor

PLOS Pathogens

Sumita Bhaduri-McIntosh

Editor-in-Chief

PLOS Pathogens

orcid.org/0000-0003-2946-9497

Michael Malim

Editor-in-Chief

PLOS Pathogens

orcid.org/0000-0002-7699-2064
---

## [Editor Report · Acceptance letter]

Dear Dr Grad,

We are delighted to inform you that your manuscript, "Diversity and evolution of a phase-variable multi-locus antigen in Neisseria gonorrhoeae," has been formally accepted for publication in PLOS Pathogens.

Best regards,

Sumita Bhaduri-McIntosh

Editor-in-Chief

PLOS Pathogens

orcid.org/0000-0003-2946-9497

Michael Malim

Editor-in-Chief

PLOS Pathogens

orcid.org/0000-0002-7699-2064